# Development of damage curves for buildings near La Rochelle during Storm Xynthia based on insurance claims and hydrodynamic simulations

Manuel Andres Diaz Loaiza[1,7], Jeremy David Bricker[1,6], Remi Meynadier[2], Trang Duong[3,4,5], Rosh Ranasinghe[3,4,5] and Sebastiaan Nicolaas Jonkman[1]

[1]Department of Hydraulic Engineering, Delft University of Technology, Delft, The Netherlands
[2]AXA Insurance, Group Risk Management, Paris, France
[3]Department of Coastal & Urban Risk & Resilience, IHE Delft Institute for Water Education, P.O. Box 3015, 2601 DA Delft, The Netherlands
[4]Department of Water Engineering & Management, University of Twente, P.O. Box 217, 7500 AE Enschede, The Netherlands
[5]Harbour, Coastal and Offshore Engineering, Deltares, P.O. Box 177, 2600 MH Delft, The Netherlands
[6]Department of Civil & Environmental Engineering, University of Michigan, Ann Arbor, MI, USA
[7]JBA Consulting, Dublin, Ireland.

*Correspondence to*: Jeremy Bricker (j.d.bricker@tudelft.nl)

**Abstract.** The Delft3D hydrodynamic and wave model is used to hindcast the storm surge and waves that impacted La Rochelle, France and the surrounding area (Aytré, Châtelaillon-Plage, Yves, Fouras and Ille du Re) during Storm Xynthia. These models are validated against tide and wave measurements. The models then estimate the footprint of flow depth, speed, unit discharge, flow momentum flux, significant wave height, wave energy flux, total water depth (flow depth plus wave height), and total (flow plus wave) force at the locations of damaged buildings for which insurance claims data are available. Correlation of the hydrodynamic and wave results with the claims data generates building damage functions. These damage functions are shown to be sensitive to the topography data used in the simulation, as well as the hydrodynamic or wave forcing parameter chosen for the correlation. The most robust damage functions result from highly accurate topographic data, and are correlated with water depth or total (flow plus wave) force.

## 1 Introduction

In February 2010 the Xynthia extratropical storm caused damage and casualties along the Atlantic coast of Spain and France (Slomp et al., 2010, Chauveau et al. 2011). The strong winds and low atmospheric pressure together with the landfall of the storm at high spring tide, generated unprecedented water levels at La Rochelle and surroundings (Bertin et al., 2014.) The present paper develops damage curves for buildings in the area where the storm surge and waves from the Xynthia storm caused the most damage. We draw on methods used to quantify damage due to hurricanes and tsunamis in the USA and Japan (Suppasri 2013, Hatzikyriakou et al., 2018, Tomiczek et al., 2017), but for the first time apply these to modern masonry

structures in Europe affected by storm surge and waves from an extratropical cyclone. Therefore, the main objective of the present study is to develop damage functions from insurance claims data supported by hydrodynamic modelling. A total of 423 reported claims in the area of study were used (Figure 1). The damage ratio (DR) is defined as the ratio of damages claimed

by each property, to the total insured value of that property. More than 9% of the structures had a damage ratio (DR) higher than 0.5 (considerable damages), 30% had DR higher than 0.2 (medium damages) and 49% had low damages. This is a typical distribution for damage claims (see for example Fuchs, S. et al., 2019).

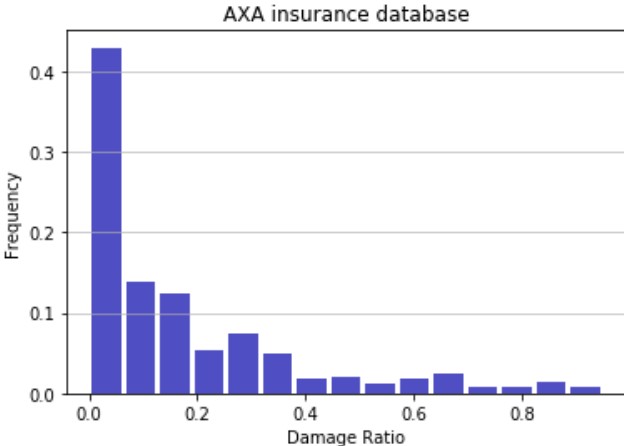

**Figure 1: Damage ratio histogram for insurance claims data in the region.**

The damage curve is an important tool in risk assessment science related to the vulnerability of structures (Pistrika et al., 2010; Englhardt et al., 2019). From the structural point of view, damage curves depend on the construction materials that buildings are made of (Huizinga, et al., 2017; Postacchini et al., 2019; Masoomi et al., 2019). Damage curves also depend on construction methods, codes, and building layout, including the distance between buildings (Suppasri et al., 2013; Jansen et al., 2020; Masoomi et al., 2019). The current paper focuses on 1-2 story masonry buildings under the effect of storm surge and wave

forces produced by an extratropical storm in northwest France. The Xynthia storm provided a rare dataset of empirical measured damage from coastal flooding in a European country. Similar analysis of damage from other storms with different return periods in the same region would help to reduce uncertainty (Breilh et al. 2014 and Bulteau et al. 2015), but for now no other claims data are available.

In flood risk assessment, the relation between damage and hazard is quantified by fragility curves and damage curves. The difference between these two is that fragility curves express the probability that a structure is damaged to a specified structural state (Tsubaki R. et al., 2016), while damage curves instead assess the cost of damage incurred by flooding of a given structure (Englhardt J. et al., 2019 and Huizinga, J., et al., 2017). For both cases it is important to highlight the fact that these curves usually rely on the flood depth alone to quantify the hazard (Pregnolato M. et al., 2015), while there are fewer studies that

attempt to represent the hazard by other quantities like the flow velocity, significant wave height or wave force (Kreibich H.

et al., 2009 and De Risi R. et al., 2017). For instance, Tomiczek T. et al. (2017) related the flow velocity to the structure damage state (DS) in New Jersey for hurricane Sandy. In the present study we relate eight different hydrodynamic variables to the damage ratio coming from insurance claims following extratropical storm Xynthia.

Damage curves are commonly developed by the correlation of field or laboratory measurements of damage, with numerical simulations of hazard level. Tsubaki et al. (2016) measured railway embankment and ballast scour in the field, and correlated this damage with flood overflow surcharge calculated by a hydrodynamic flood simulation. Englhardt et al. (2019) and Huizinga et al. (2017) used big-data analytics to correlate tabulated damages with estimated flood levels over a large scale. Pregnolato et al. (2015) showed that most damage functions are based on flood depth alone, though a few also consider flow

speed (De Risi et al., 2017; Jansen et al., 2020) or flood duration. The water depth is an important variable since it accounts for the static forces that act on a structure. Nevertheless, in storm events, structures close to the coast at a foreshore/backshore can be subjected to dynamical forces like the action of flow and waves (Kreibich et al., 2009; Tomiczek et al., 2017). For this reason, in order to consider other possible forces the following hydrodynamic parameters are analysed: water depth ($h$), flow speed ($v$), unit discharge ($hv$), flow momentum flux ($\rho h v^2$), significant wave height ($H_{sig}$), total water depth ($h + H_{sig}$), wave

energy flux ($E_f$), and total force ($\frac{E_f}{C_g} + \rho h v^2$). The wave energy flux is defined via Eq. (1) as in Bricker et al. (2017).

$$E_f = \frac{1}{16}\rho g H_{sig}^2 C_g, \tag{1}$$

where $H_{sig}$ (m) is the significant wave height, $C_g$ (m/s) is the wave group velocity, $\rho$ (kg/m$^3$) is the water density and $g$ (m/s$^2$) is the acceleration due to gravity, and $C_g = \sqrt{gh}$ over land where waves impact buildings.

**2 Methods**

Damage curves were developed by hindcasting the hazard with a meteorological model, followed by a hydrodynamic (tides and storm surge) and wave model, and then correlating the resulting flood conditions with claimed damages (Fig. 2).

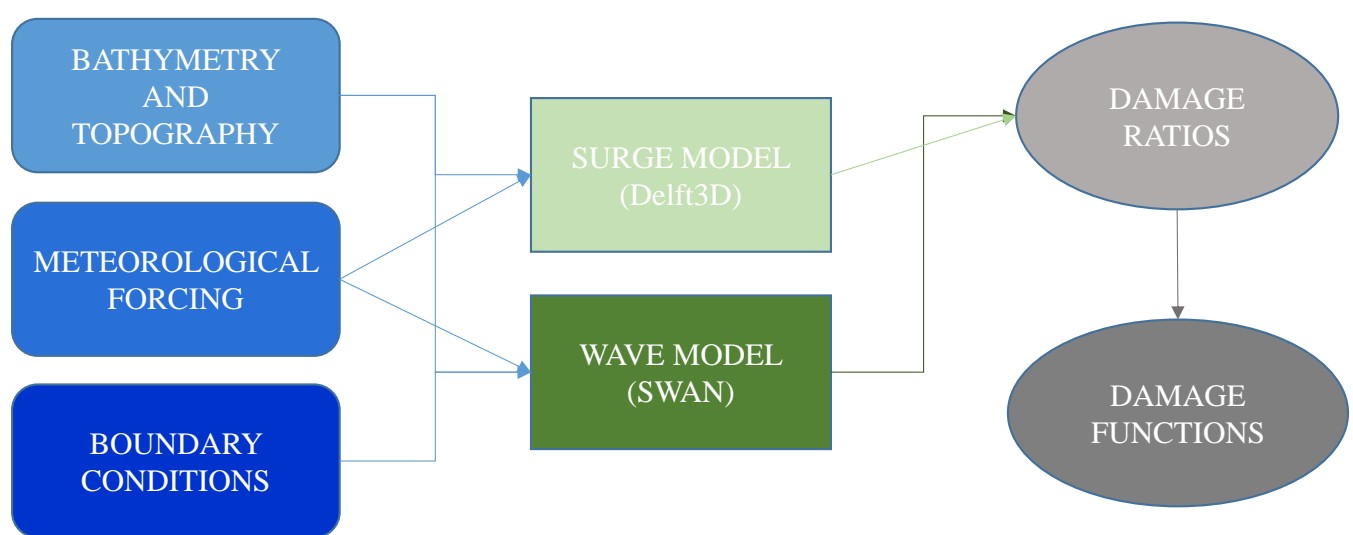

**Figure 2: Flow chart of the framework used in development of damage curves.**

### 2.1 Meteorological model setup and description of the Xynthia storm

From 23 February 2010 Meteo France start recording a low pressure front that was forming in the north Atlantic and passed north of Spain on 27 February, with a minimum pressure of 966 hPa (Fig. 3). Early in the morning of 28 February it made landfall on the French Coast at the same time that a high astronomical tide was developing, causing a total of 65 casualties in the regions of Vendee and Charente-Maritime and approximately 2.5 billion euros damage to agricultural (oyster farms and mussels) infrastructure, the tourist industry and residential/commercial zones  (Slomp et al., 2010). To generate pressure and

wind fields to drive the storm surge model, dynamically downscaled surface meteorological data were generated for the French Atlantic study region (Figure 3). This contains zonal and meridional winds 10 m above ground (u10, v10) and surface pressures over sea and land, with 3.5 km spatial resolution and 3 hrs temporal resolution. The dynamical downscaling was performed with the regional climate model WRF (Skamarock et al., 2008), based on NCEP CFSR renalaysis data (Saha et al., 2010). The regional non-hydrostatic WRF model (version 3.4) simulated 15 February 2010 until 05 March 2010. The initial and lateral

boundary conditions are taken from the CFSR reanalysis at 0.5° resolution, updated every 6 h. The horizontal resolution is 7 km; we use a vertical resolution of 35 sigma levels with a top-of-atmosphere at 50hPa. The simulation domain was chosen to be wide enough in latitude and longitude for WRF to fully simulate the large-scale atmospheric features of the Xynthia extratropical cyclone. A spin-up time of 5 days was considered in the study to remove spurious effects of the top layer soil moisture adjustment even though most of the analyses here are performed over the ocean. Land surface processes are resolved

by using the NOAA Land Surface Model scheme with four soil layers. Numerical schemes used in the WRF simulation to downscale Xynthia data are the Multi-Scale Kain-Fritsch scheme for convection, the Yonsei University scheme for the planetary boundary layer, the WRF Single-Moment 6-class scheme for microphysics, and the RRTMG scheme for shortwave and longwave radiation. WRF outputs are generated every 3 hours.

## 2.2 Hydrodynamic model of the Xynthia Storm

In order to capture the hydrodynamic storm characteristics, a regional model domain over the Atlantic Spanish and French coasts was built. As shown schematically in Fig. 2, Delft3D calculates non-steady flow phenomena that result from tidal and meteorological forcing on a rectilinear or a curvilinear grid (Deltares, 2021). At the same time, and coupled with Delft3D, a spectral wave model (SWAN) calculates significant wave height and period fields. Delft3D and SWAN were used to hindcast the physical forcing at the locations of all claims in the database. Afterwards, a probability standardized normal distribution

function as proposed by Suppasri et al. (2013) was used to develop damage curves by correlating claimed damage with a variety of hydrodynamic forcing variables. To conserve computational resources and reduce computation time, domain decomposition (2-way hydrodynamic nesting) was implemented with grids of resolution of ~2km over the open ocean, ~400m close to the study area and ~80m over the area of claims data (Figure 3).

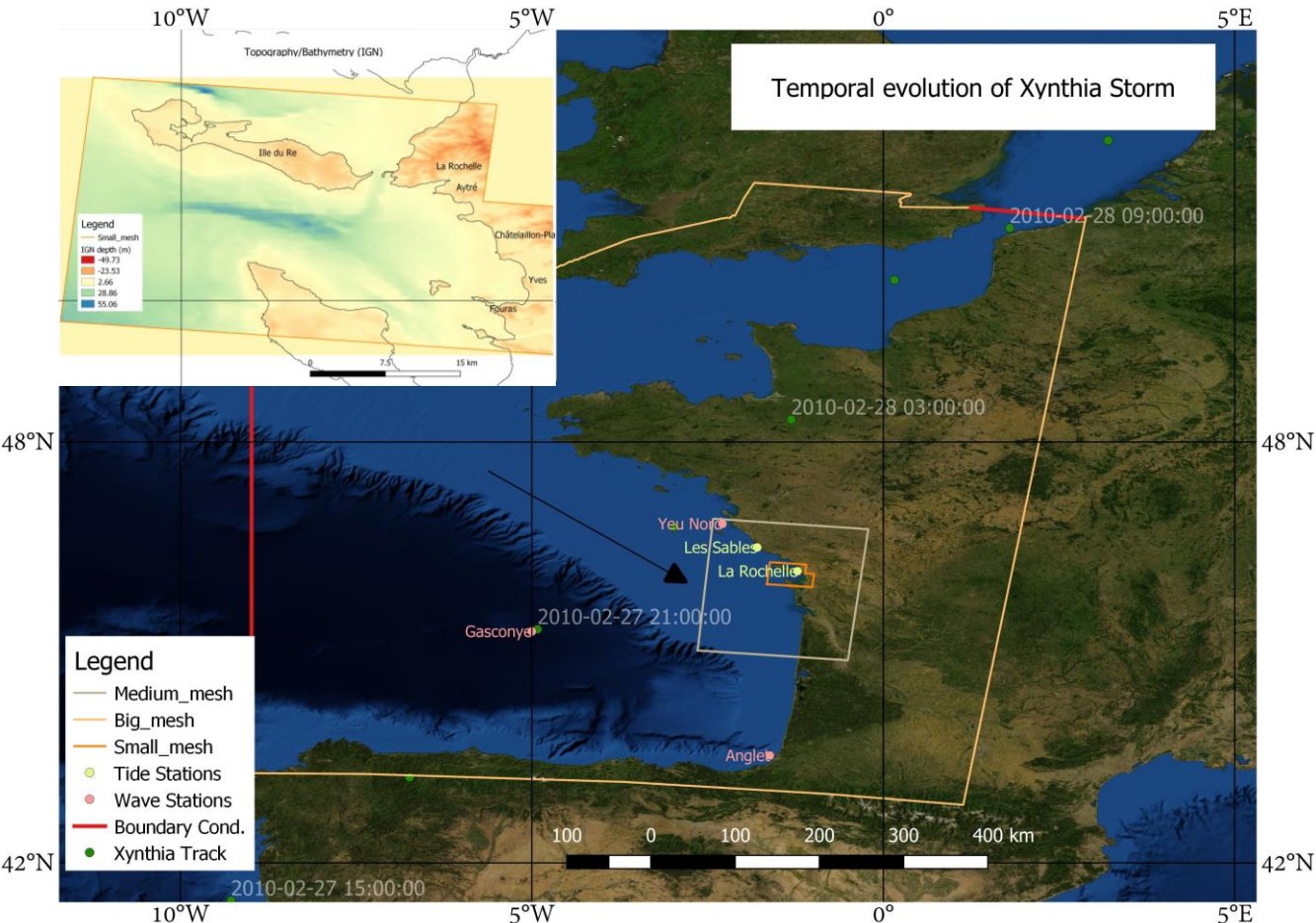

**Figure 3: Domain decomposition with three nested grids running in parallel. The center of the Xynthia storm is shown as a triangle at the time of minimum atmospheric pressure of 966 hPa at 2010-02-27 21:00:00 (Extreme Wind Storm Catalogue). Topographic map inset covers the smallest domain shown on the large map. Satellite image by OpenLayers – QGIS.**

### 2.2.1 Topography and Bathymetry

We use two types of topography datasets: a global dataset for the bathymetry/topography (GEBCO 2019, which is based on
SRTM 15+ v2 over land), and a higher resolution bathymetry (MNT – HOMONIM project) and topography (IGN institute).
Additionally, a survey of flood wall height was performed during August 2020 in order to include flood walls as thin weirs
inside the Delft3D model, and in this way overcome the fact that inside the high resolution 5m topography, these structures
are not represented, as suggested by Bertin et al. (2014). Luppichini et al. (2019) and Ettritcha et al. (2018) found that the
quality of bathymetry and topography data has a large effect on estimation of the hazard, and Brussee et al. (2021) similarly
found topography data quality affects resulting damage estimates. In order to investigate the effect of the quality of topographic
and bathymetric data on the resulting damage functions, three scenarios are considered in our work (Table 1).

**Table 1: Case studies for investigating sensitivity of model result to DEM resolution.**

| Item | Low resolution (a) | High resolution (b) | High resolution + structures (c) |
|---|---|---|---|
| Topography | GEBCO (500m) | IGN (5m) | IGN (5m) + flood walls surveyed by the authors with an RTK-GPS |
| Bathymetry | GEBCO (500m) | GEBCO (500m) in deep water + MNT (100m) nearshore | GEBCO (500m) in deep water + MNT (100m) nearshore |

## 2.3 Hydrodynamic and Wave Model setup

Delft3D was coupled together with SWAN in order to hindcast storm tide and waves. Model boundary conditions consisted
of astronomical tidal water elevations from the Global Tide and Surge Model (GTSM) of Muis et al. (2016) for the period
from 20 February until 1 March 2010. The hydrodynamic model was run with a computational time step of 30 sec and a
uniform Manning's n of 0.025. The air-sea drag coefficient of Smith and Banke (1975) was used. Other model parameters
retained their default settings.

### 2.4 Hydrodynamic and wave model validation

### 2.4.1 Storm tide validation

The hydrodynamic model was run from 20 February until 1 March 2010, the duration of the meteorological forcing data, with
GTSM astronomical tide boundary conditions. For validation, three accuracy indicators are assessed: root mean square error
(RMSE, Equation 3), relative root square error (RRSE, Equation 4), and the Pearson correlation coefficient (ρ, Equation 5).

$$RMSE = \sqrt{\frac{\Sigma_1^T (y'-y)^2}{T}}, \qquad\qquad (3)$$

$$RRSE = \sqrt{\frac{\Sigma_1^T (y'-y)^2}{\Sigma_1^T (y-\bar{y})^2}}, \quad \bar{y} = \frac{\Sigma_1^T y}{T} \tag{4}$$

$$\rho_{y,y'} = \frac{cov(y,y')}{\sigma_y \sigma_{y'}}, \tag{5}$$

where $y'$ is the predicted value, $y$ is the actual value and $\bar{y}$ is the average of the actual values to predict, $T$ is the number of values, and $\sigma$ indicates the standard deviation

After 2 days of model spin-up (the time required for the model to correct the assigned initial condition), the comparison between the observed water levels from SHOM-Coriolis tide gauges (http://www.coriolis.eu.org/), and modelled water levels from Delft3D, during the whole simulation is acceptable (Fig. 4) according to the results for the goodness of fit indices in table 2. If we compare these values with typical values in the literature such as Matte et al. (2014) or Tranchant et al. (2021) we observe the current modelled water levels fit the observations well. Note that the Les Sables gauge failed at the peak of the storm (on 2010-02-28 03:00:00) so a data point is missing in the observations at that time. At La Rochelle the difference between the observed and modelled water level is only 36cm at peak storm tide.

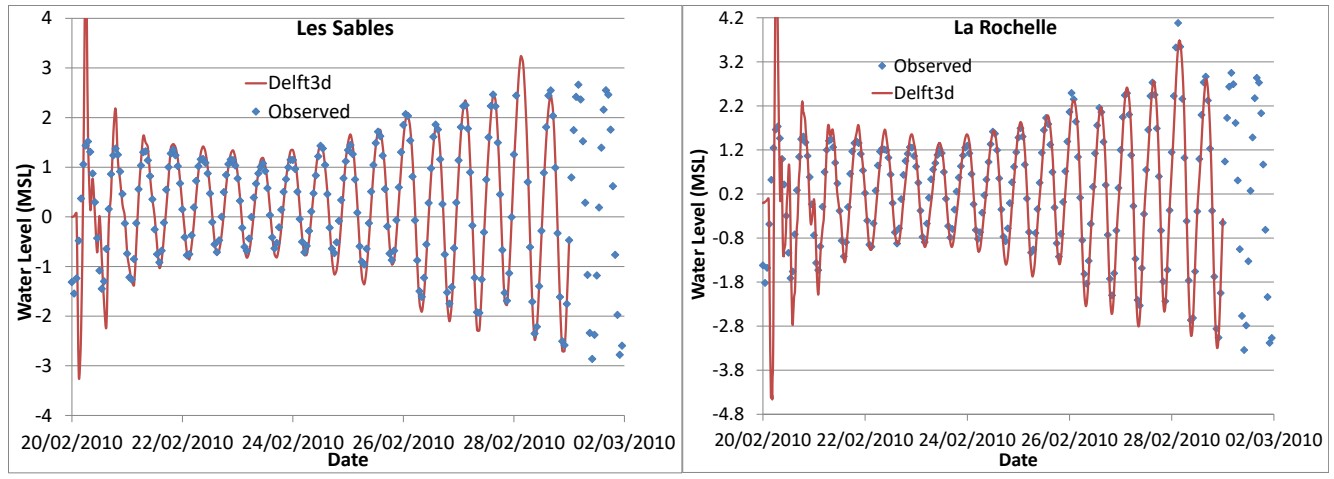

**Figure 4: Observed and modelled tide at La Rochelle and Les Sables. Note that during the peak of the storm tide at Les Sables, the tide measuring gauge was out of operation, resulting in a missing data point in that data series.**

### 2.3.2 Wave model validation

The wave model was validated against data from the SHOM-Coriolis operational oceanography center (http://www.coriolis.eu.org/About-Coriolis) in Figure 5. Important to mention is that the data available at the buoys stations do not include the significant wave height, therefore the swell height was extracted to compare the results from Delft3D-SWAN. The uncertainty produced by the meteorological downscaling by means of the WRF model in the hindcast of the winds can add errors in the results. Unfortunately, no more meteorological information is available. If we again compare the indices

from table 2 to those found in the literature such as Baron-Hyppolite et al. (2019), we find comparable goodness of fit between modelled and measured waves.

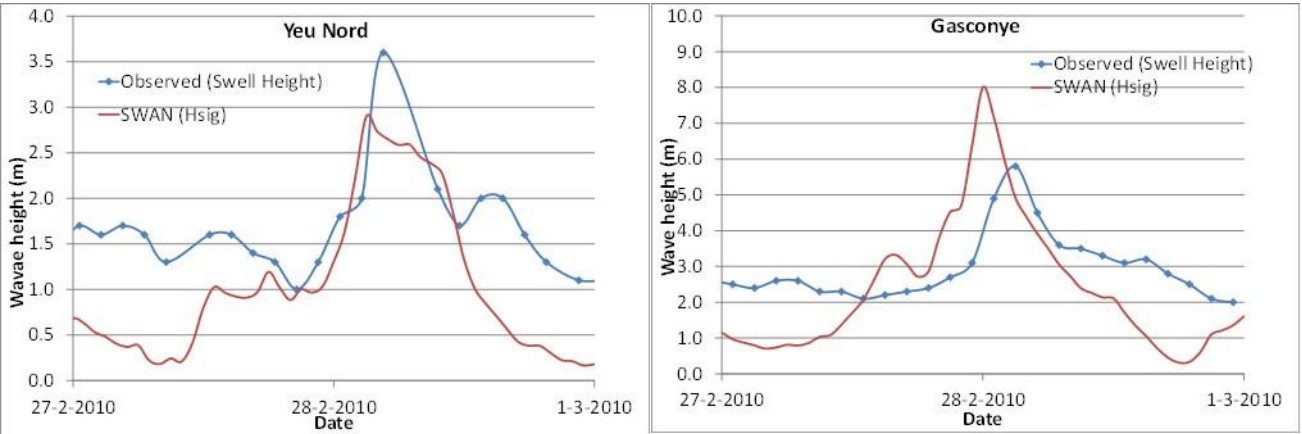

**Figure 5: Deep water buoys of Yeu Nord (left) and Gasconye (right). In the first case the buoy is located close by an Island of the same name. The second is located in the open ocean almost in the middle of the Bay of Biscay.**

**Table 2: Goodness of fit for water level and wave measurements compared with the results of Delft3D-SWAN.**

| Station | RMSE (m) | $\rho$ | RRMSE |
|---|---|---|---|
| Gazconye | 1.5434 | 0.6228 | 0.10679 |
| Yeu Nord | 0.8668 | 0.6985 | 0.1116 |
| Les Sables | 0.4959 | 0.9197 | 0.1381 |
| La Rochelle | 0.4991 | 0.9157 | 0.1374 |

## 2.5 Damage curves

Damage curves express the amount of damage experienced by a structure, relative to the structure's total insured value. The cumulative distribution function, in terms of the standardized normal distribution function with the damages (Suppasri et al., 2013; Thapa et al., 2020; Sihombing and Torbol, 2016) is shown in Equation (2).

$$P(x) = \Phi\left[\frac{x-\mu}{\sigma}\right], \tag{2}$$

where $P(x)$ is the cumulative probability of the damage ratio with values between 0 and 1, and $x$ is the hydrodynamic variable, $\Phi$ is the standardized normal distribution, $\mu$ is the median and $\sigma$ the standard deviation (Tsubaki et al., 2016). It is also very common to express equation (1) as a logarithmic function in order to obtain easily the parameters of the distribution with least square fitting as proposed by Suppasri et al. (2013). In the present paper, the parameters are assessed using the L-moments

package within the open source program R. In this way, it is possible to relate different hydrodynamic variables with the damage ratio. From the 423 claims data within our domain, approximately 185 are on Ille du Re, and the remaining 238 in the towns of La Rochelle, Aytré, Yves, Châtelaillon-Plage and Fouras. At each claim location, the maximum of each hydrodynamic variable was extracted and from this the damage curves were compiled.

**3 Results**

After determining the model hydrodynamic and wave results (Fig. 6) at the location of each claim location, the data were subdivided into ten categories according to damage ratio level, and box-whisker plots were built to display the entire dataset and analyse the trend of the data (Appendix A). Among the flow-only variables, the unit discharge ($hv$) appears to have the clearest trend and least scatter. From the variables related to both flow and waves, the total force ($\frac{E_f}{c_g} + \rho h v^2$) appears to have

the clearest trend and correlation with the damage ratio.

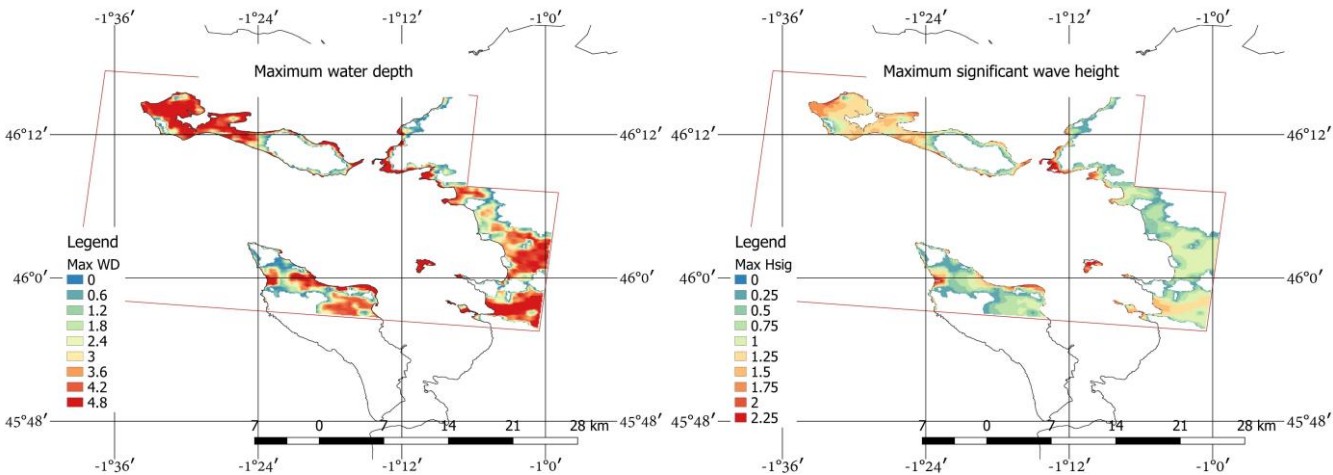

**Figure 6: Maximum water level and maximum significant wave height footprints for the small model domain (case study area).**
**Water depth and wave height are in units of m. The purple rectangle indicates the limits of the small domain, outside of which data is not shown.**

## 3.4 Damage curves from each digital elevation model

In order to build damage curves with equation (2), the median values are extracted from the boxplots of appendix A (figures A1 to A3) for each variable. In Fig. 7

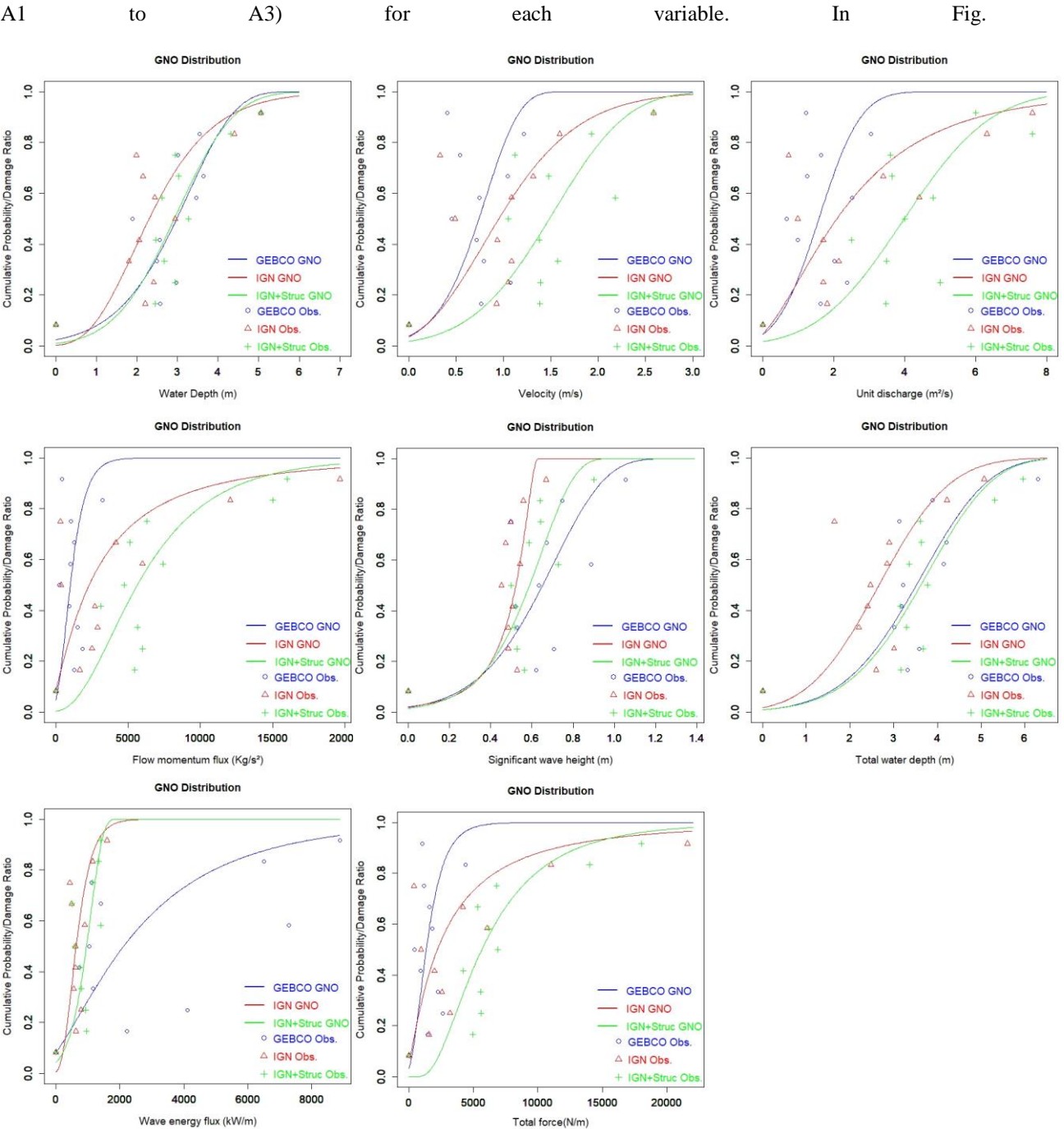


Figure 7 the damage curves for each hydrodynamic parameter are displayed as 3 lines, one for each digital elevation model of Table 1. Similar to Reese and Ramsay (2010), we find that greater than 90% of damage occurs in the first 5m of flood depth.

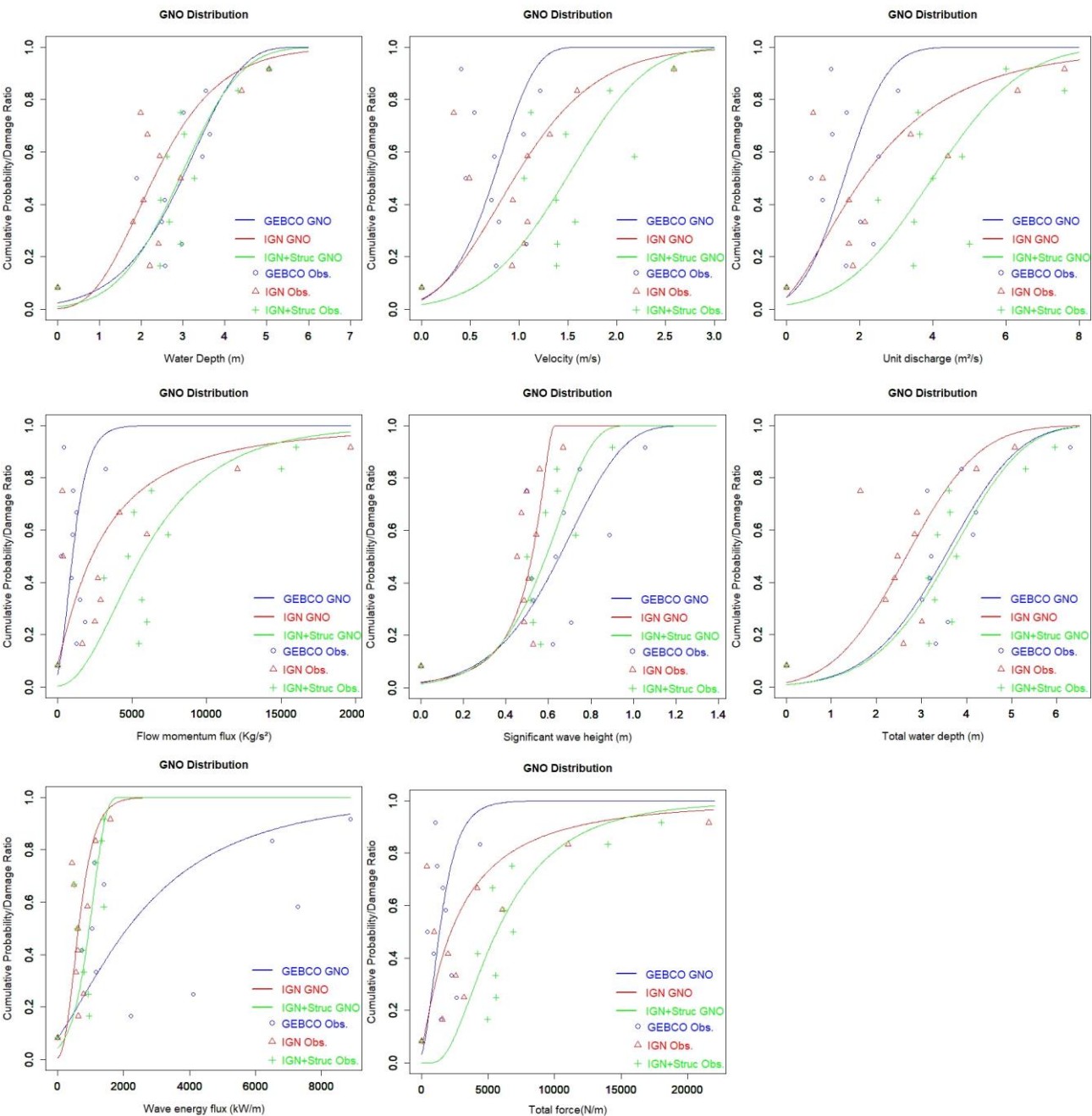

**Figure 7: Damage curves for the surge and wave variables ( $h, v, hv, \rho hv^2, H_{sig}, h + H_{sig}, E_f, \frac{E_f}{C_g} + \rho hv^2$ ), and different bathymetry/topography conditions (Table 1). Markers indicate the observed data and lines the fitted statistical distributions.**


Table 3 shows that among the hydrodynamic parameters related only to storm surge, the water depth best fits Equation (2), with the lowest errors (RMSE and RRSE) and the highest Pearson coefficient (ρ). Among the combined surge and wave parameters, the best correlation is the total (flow plus wave) force, using the IGN+Structures topography and bathymetry (Table3). This is related to the fact that this digital elevation model includes thin flood walls that contribute to protection, and which can substantially modify the flow and wave fields over land.

**Table 3: Goodness of fit for the flow only, and flow plus wave, parameters. The best fits for flow-only parameters are indicated in bold, and the best fits for flow plus wave parameters are indicated in bold/italic.**

| Variable | $RMSE$ (m) | | | $\rho$ | | | $RRSE$ | | |
|---|---|---|---|---|---|---|---|---|---|
| | GEBCO | IGN | IGN + Structures | GEBCO | IGN | IGN + Structures | GEBCO | IGN | IGN + Structures |
| Water depth (h) | 0.1595 | 0.1898 | **0.1495** | 0.8134 | 0.7344 | **0.8328** | 0.1009 | 0.1145 | **0.0902** |
| Flow speed (v) | 0.3586 | 0.2561 | 0.2234 | 0.1284 | 0.5387 | 0.6406 | 0.2268 | 0.1545 | 0.1347 |
| Unit discharge ($hv$) | 0.3352 | 0.2272 | 0.2120 | 0.2421 | 0.6558 | 0.6744 | 0.2120 | 0.1370 | 0.1278 |
| Flow momentum flux ($\rho h v^2$) | 0.3542 | 0.2540 | 0.1822 | 0.1314 | 0.5759 | 0.7622 | 0.2136 | 0.1532 | 0.1099 |
| Significant wave height ($H_{sig}$) | 0.2211 | 0.2030 | 0.1600 | 0.6432 | 0.6901 | 0.8066 | 0.1398 | 0.1224 | 0.0965 |
| Total water depth ($h + H_{sig}$) | 0.1767 | 0.2217 | 0.1522 | 0.7575 | 0.6404 | 0.8265 | 0.1117 | 0.1337 | 0.0918 |
| Wave energy flux ($E_f$) | 0.2649 | 0.2391 | 0.2307 | 0.5519 | 0.5851 | 0.6510 | 0.1676 | 0.1442 | 0.1391 |
| Total force ($\frac{E_f}{C_g} + \rho h v^2$) | 0.3307 | 0.2494 | *0.1499* | 0.2396 | 0.5888 | *0.8387* | 0.2092 | 0.1504 | *0.0904* |

In Appendix B a comparison to other 2 typical distribution functions is carried out. It can be seen that the Gamma and GNO distributions have similar goodness of fit indicators, while the Log Normal distribution performs slightly worse overall. An analysis on the uncertainty due to the statistical distribution selection or the inclusion of properties with no damages can be found in Fuchs et al. (2019). Another source of uncertainty, in addition of the selected statistical distribution, is the parameter fitting method (Diaz-Loaiza, 2015). Typical methods for this purpose include L-moments, maximum spacing estimation, maximum likelihood, moment method, and least squares method (Oosterbaan, 1994).

## 4 Discussion

The present paper considered the influence of flow-only variables ($h, v, hv, \rho h v^2$), and combined flow-wave parameters ($Hsig$, $h + Hsig$, $Ef$, $\frac{Ef}{Cd} + \rho h v^2$). Flow depth and total (flow plus wave) force produce the best fits with analytical functions.

Goodness of fit to damage curves improves with quality of the topographic data used (Table 1). However, when applying
damage curves in practice, it is important to base predictions off a similar model setup to that used when calculating the damage
curves in the first place (Brussee et al., 2021). For example, if damage curves are built using coarse topography that neglects
the presence of thin seawalls (i.e. sheetpile/cantilever walls, or T- or L- walls), then the buildings protected by these walls
might experience more intense hydrodynamic conditions in the simulation than if the walls had been present in the simulation.
Since the actual recorded damage does not depend on the model used to calculate the hydrodynamic forcing conditions, damage
curves developed using the coarse resolution topography will be shifted to the right relative to damage curves generated with
the thin floodwalls present. If these damage curves generated using a coarse resolution simulation are then applied for damage
prediction by an external user who applies a high resolution simulation that resolves floodwalls, the reduced forcing (due to
the presence of these floodwalls) will generate a non-conservative result (too little damage), because the damage curves had
been generated using forcing data from a simulation where the floodwalls had not been present. Therefore, when damage
curves are reported in the literature, it is important to quantify how these vary with the topography used in the simulations on
which the damage curves are based. However, in the current paper, Fig. 7 shows that damage curves do not vary consistently
leftward or rightward as topographic data are improved. This is because the response of forcing to the presence of these walls
is more complex than simply reducing wave height. If not overflowed, walls reduce damage greatly. However, water depth
can be exacerbated in front of walls, and flow can be channelled and intensified along walls, all increasing hydrodynamic
forcing in some locations, preventing a simple relation between topographic resolution and damage curve robustness.

In addition to the general sensitivity of damage curves to topographic data quality, the damage curves displayed in Figure 7
do not consider certain physical wave-driven phenomena such as wave overtopping of structures (Lashley et al., 2020a; Ke et
al., 2021) or infragravity waves generated by waves breaking in shallow water (Roeber and Bricker, 2015). For instance
Lashley et al. (2019) discussed the importance of dike overtopping due to infragravity waves on nearshore developments that
can induce wave-driven coastal inundation. The wave model used here, SWAN, does not include infragravity waves, nor does
the combined Delft3D/SWAN flow/wave model simulate wave overtopping of dikes, possibly leading to an underestimation
of the hydrodynamic forces on buildings, which would affect the resulting damage functions. However, consideration of wave
overtopping and infragravity effects requires either phase-resolving wave simulations or empirical relations specific to the
local topography (Lashley et al., 2020b), though this is beyond the scope of the current study, and is similarly neglected by
most other large-scale inundation studies (i.e., Sebastian et al, 2014; Kress et al., 2016: Kowaleski et al., 2020). Nonetheless,
the effect of infragravity oscillations and wave overtopping on resulting damage is an important item for future research.

Another important factor mentioned by Bertin et al. (2015) was the particular track direction of the storm that for the Xynthia
event induced a young sea state, enhancing the surface stress, and adding up to 40 cm to the theoretical surge and tide of their
model. The uncertainty and variability within this methodology can be explained by two factors: 1) the hydrodynamic
modelling, and consequently, uncertainty in the hydrodynamic variables, and 2) uncertainty in the claims data. Regarding the

first point, there is a trend that indicates that better topography/bathymetry data gives hydrodynamic variables that correlate better with the damage ratio. This is because higher resolution data generally more accurately reproduces the actual flood conditions (Luppichini et al., 2019 and Ettritcha et al., 2018). Damage curves developed with a better representation of the topography (IGN + structures) improve the accuracy indicators (Table 3), though scatter in the data itself (Figures A1, A2 or A3) is large for all topographies. This first point is also related to the mesh resolution and the roughness coefficients used. The second point deals with the quality of the damage ratio data. It is known that insurance claims can sometimes be subject to fraud or information distortion. In addition, variables related to the vulnerability of the assets like the construction characteristics, the materials, the quality and the age of the structures (Paprotny et al., 2021) play important roles in determining whether a particular hydrodynamic is related to damage. This adds a degree of complexity to the analysis. Finally, it is worth mentioning that if more detailed information from the claims data is available (like structure type, number of floors, damage stage etc.) then a more detailed fragility functions can be generated instead of the bulk damage functions determined here.

## 5 Conclusions

Insurance claims data facilitated generation of damage curves for structures located in La Rochelle and surroundings. This provides valuable information for predicting future damages that can be expected from an extratropical storm strike on the French Atlantic coast. In the present study, the hydrodynamic variables that correlated best with the damage ratio are the flow depth and the total (flow plus wave) force for the flow-only and flow-plus-wave-related variables respectively. In addition to the sensitivity of results to resolution of the topographic and bathymetric data, the inclusion of thin flood walls via a land survey carried out by the authors also had a significant effect on the damage functions generated. This is important to note, as thin steel or concrete structures like flood walls are typically only a few decimetres thick, and therefore do not appear in digital elevation models. The effect of these thin structures on the resulting damage functions shows the importance of locally sourcing elevation data for the thin structures that are present when conducting risk analyses for coastal regions. However it is imperative to keep in mind agreement between the simulations used for developing the damage relations in the first place, with those where the damage relations are applied for further risk analysis.

**Data availability**

Data will be available after publication on the data repository of TU-Delft at https://data.4tu.nl/search as AXA Xynthia storm research project.

**Author contributions**

The present paper is based on the postdoctoral research of Manuel Andres Diaz Loaiza on the research project INFRA. MADL, JB, RM, TD and RR conceptualized the study and maintained meetings of progress for the project. MADL made the calculations and wrote the document.  JB and SJ provided repeated feedback on the manuscript.

**Author contributions**

The contact author has declared that neither they nor their co-authors have any competing interests.

**Acknowledgements**

This work is funded by the AXA Joint Research Initiative (JRI) project INFRA: Integrated Flood Risk Assessment. A special acknowledgement to Adri Mourits from Deltares for the help provided with the Delft3D debugging and Christopher Lashley for the help during the field trip in Ille du Re and surrounding during August 2020.

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

## Appendix A

Whisker plots from which damage curves are developed are shown in Figures A1, A2, and A3. Digital Elevation Models are as described in Table 1. The damage curves of Fig. 7 use the median values (red lines) from each of the figures in this appendix.

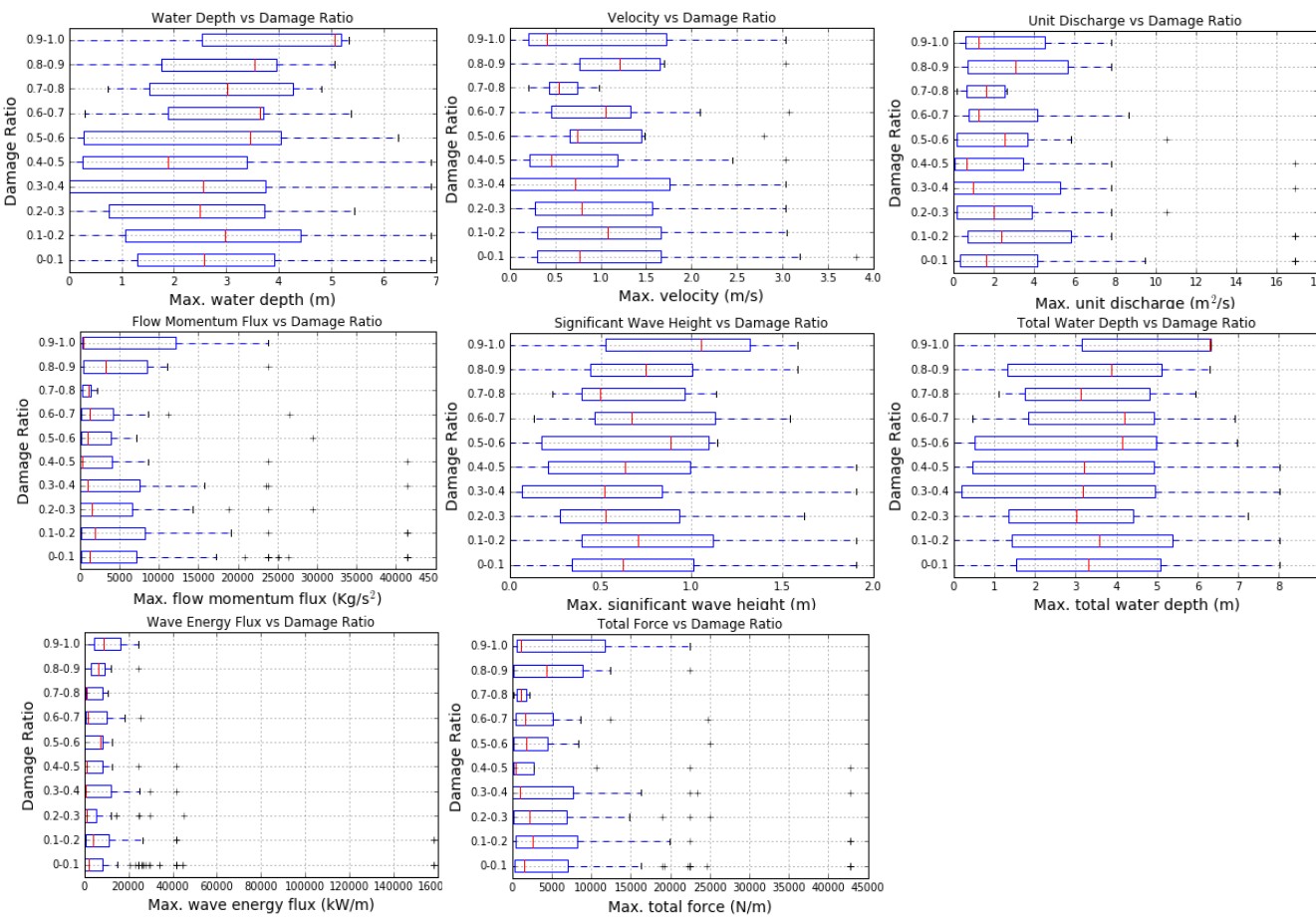


**Figure A1: Box-whisker plots for the variables ($h, v, hv, \rho h v^2, H_{sig}, h + H_{sig}, E_f, \frac{E_f}{c_g} + \rho h v^2$) with the GEBCO DEM.**

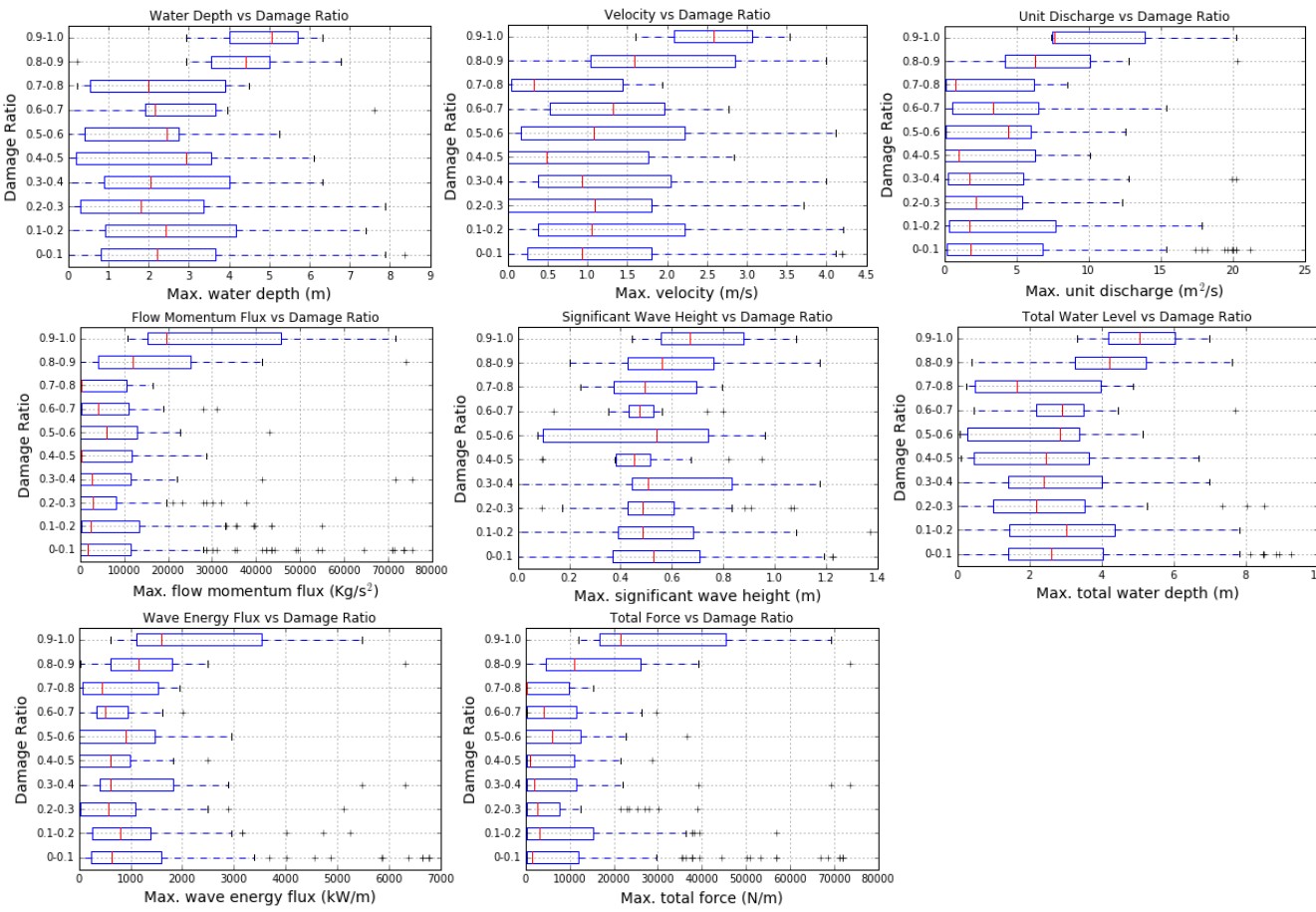

Figure A2: Box-whisker plots for the variables ($h, v, hv, \rho h v^2, H_{sig}, h + H_{sig},\ E_f,\ \frac{E_f}{C_g} + \rho h v^2$) with the IGN DEM.


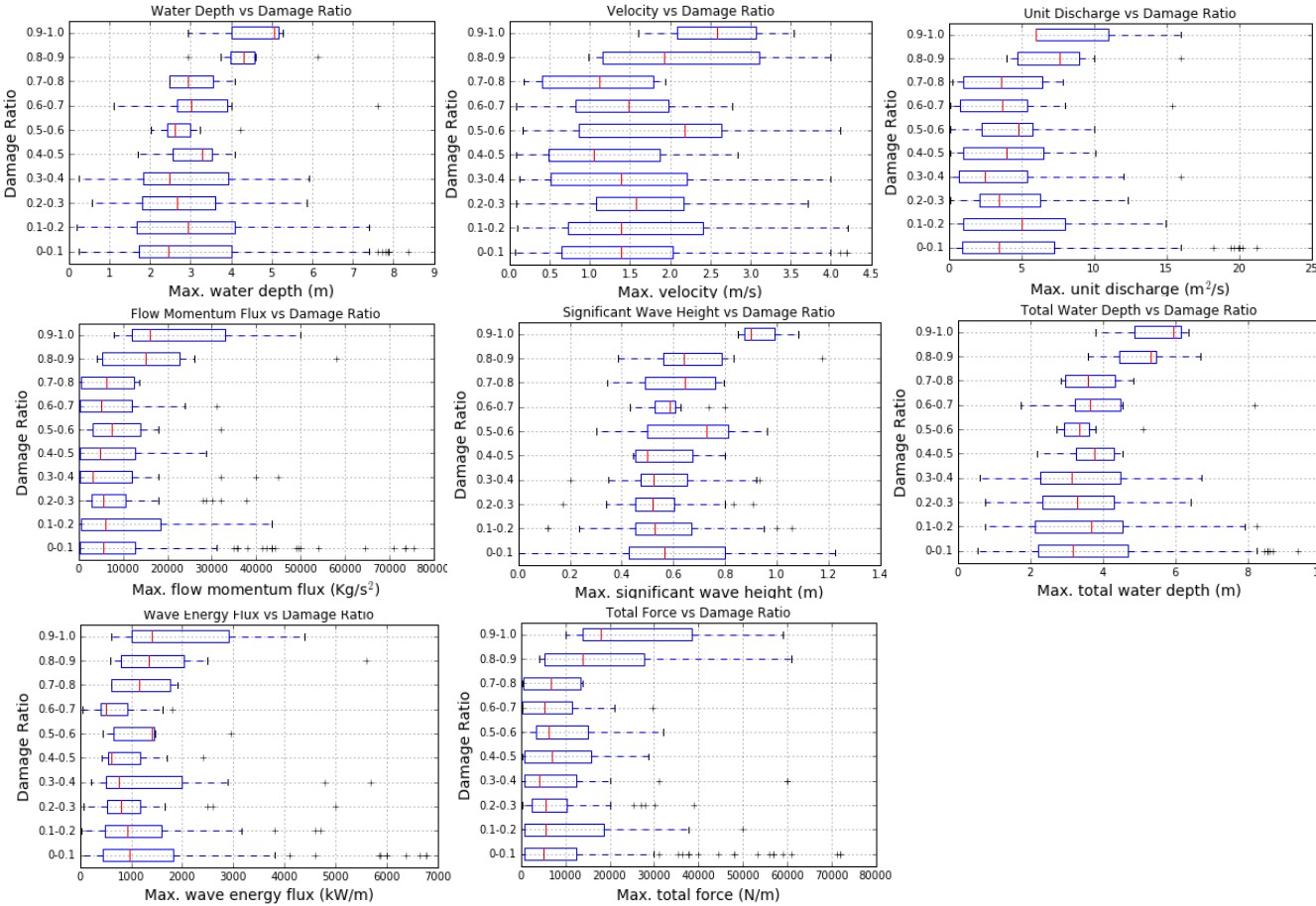

Figure A3: Box-whisker plots for the variables ($h, v, hv, \rho h v^2, H_{sig}, h + H_{sig}, \ E_f, \frac{E_f}{c_g} + \rho h v^2$) with the IGN+Structures DEM.



Appendix B

Probability distribution comparison for the bathymetry/topography of IGN+structures.

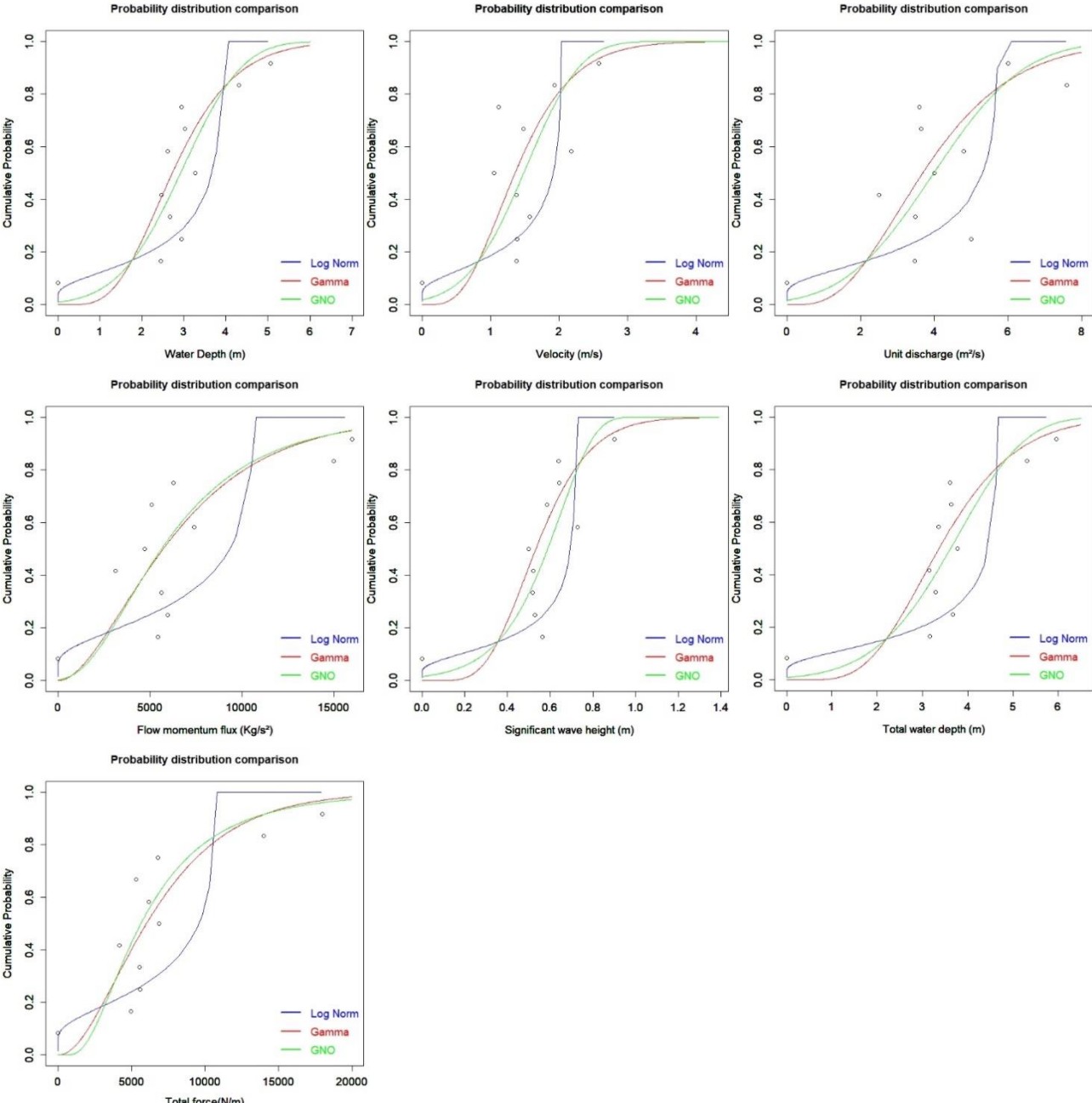

**Figure B1: Comparison of three typical statistical distributions used for damage function development. The points correspond to the observed data and lines for different statistical distributions.**

**Table B1: Goodness of fit indices for the Gamma, Log Normal and Generalized Normal statistical distributions. The best fits for flow-only parameters are indicated in bold, and the best fits for flow plus wave parameters are indicated in bold/italic.**

| Variable | *RMSE* (m) | | | $\rho$ | | | *RRSE* | | |
|---|---|---|---|---|---|---|---|---|---|
| | Gamma | Log Nor | GNO | Gamma | Log Nor | GNO | Gamma | Log Nor | GNO |
| Water depth ($h$) | 0.1574 | 0.2290 | **0.1495** | 0.8256 | 0.7722 | **0.8328** | 0.0949 | 0.1381 | **0.0902** |
| Flow speed ($v$) | 0.2306 | 0.2802 | **0.2234** | 0.6180 | 0.6087 | **0.6406** | 0.1390 | 0.1690 | **0.1347** |
| Unit discharge ($hv$) | 0.2150 | 0.2440 | **0.2120** | 0.6704 | **0.7244** | 0.6744 | 0.1296 | 0.1471 | **0.1278** |
| Flow momentum flux ($\rho hv^2$) | **0.1790** | 0.2341 | 0.1822 | **0.7686** | 0.7591 | 0.7622 | **0.1079** | 0.1412 | 0.1099 |
| Significant wave height ($H_{sig}$) | 0.1719 | 0.2888 | ***0.1600*** | 0.7987 | 0.6065 | ***0.8066*** | 0.1037 | 0.1742 | ***0.0965*** |
| Total water depth ($h + H_{sig}$) | 0.1604 | 0.2453 | ***0.1522*** | 0.8195 | 0.7582 | ***0.8265*** | 0.0967 | 0.1479 | ***0.0918*** |
| Wave energy flux ($E_f$) | 0.2522 | 0.2601 | ***0.2307*** | 0.5774 | ***0.7130*** | 0.6510 | 0.1521 | 0.1568 | ***0.1391*** |
| Total force ( $\frac{E_f}{c_g} + \rho hv^2$) | ***0.1462*** | 0.2318 | 0.1499 | ***0.8410*** | 0.7713 | 0.8387 | ***0.0882*** | 0.1398 | 0.0904 |