# Peer review of "Development of damage curves for buildings near La Rochelle during Storm Xynthia based on insurance claims and hydrodynamic simulations"

_Natural Hazards and Earth System Sciences, 2021_

## Author Response (AR1)

Reviewer 1

Dear authors,

This is an interesting research on damage curves development based upon insurance damage data and hydrodynamic model results. The methods are clearly presented in the paper. The validation results of hydrodynamic models seems reasonable. The only concern of myself is the proposed standard normal distribution of damage curves because the results of all the damage curves developed in this paper is based on this hypothesis. I was wondering how to validate these damage curves? How will the insurance company utilize the damage curves for further risk analysis?

*Response: Dear reviewer, thank you for your comments and your observations. I will answer them in the next paragraphs. For the concern related with the use of standard normal distribution for the damage function development we decided to use this function since is one of the most common statistical distribution used to this scope. Nevertheless, now we decide to apply another statistical distribution in order to compare with the current results. The results will be displayed in the appendix for the newer version both together with the comments of Xavier Bertin. In regards on how insurance companies can use the damages curves for future risk analysis the answer is that we are helping to determine which physical quantities (flow velocity, water depth, significant wave height etc..) generates the best correlation with damage data, and this is something insurance companies are willing to know.*

Other comments:

1. a thorough discussion of literature review is missing in the 'Introduction'. what is the common way of developing damage functions? what do the previous researchers have done? what are the main conclusions of their works? what is the current research gap? what is the scientific contribution of this research? please also explicitly explain the significance of this work.

*Response: In the beginning we decide to be the most direct with the structure and content of the paper, since the same is intended to people who already have experience in the topic. But now that this observation arise we decide to include two paragraphs on this matter. Thank you for the observation*

2. Line 56: 'In' à 'in'

*Response: yes, in the new version line 56 is changed*

3. Line 60: I suggest add the units of these parameters. e.g. Hsig is the significant wave height [m];

*Response: we will guarantee that in the tables and all the figures appear the correspondent units*

4. Line 65: Figure 23àFigure 3

*Response: yes, the figure title is changed in the text of line 65*

5.      Caption of Table 1 could be 'Description of three scenarios of topography and bathymetry data used in the model : low resolution (a), high resolution (b), high resolution + structures (c)'.

*Response: We feel the current caption is enough explicative of the content in table 1.*

6.      I suggest zoom in the study area of Ille du Re and La Rochelle to show the water depth and Hsig. Otherwise the readers cannot get useful information from Figure 6. This figure currently didn't convey clear information on water depth and wave height.

*Response: yes, a better image with a zoom over Ille du Re and La Rochelle will be included*

7.      Both figure 7 and Table 2 show that the damage curves for water depth and total water depth have good and very similar fitting curves for coarse (GEBCO) and fine (IGN+structure) data. It seems that damage curve is not that sensitive to the topography data for the variable of water depth. I recommend to discuss it in the section of 'Discussion'.

*Response: yes, that behave was initially detected although the real explanation for this is not clearly understood. Although only IGN for this variable have worst RRSE, RMSE and Pearson coefficient compare to IGN+Structures and GEBCO, these values are good in between the rest of the variables, indicating Water Depth and Total Water are good descriptive variables for the damage curves in this case study. Nonetheless, for all the variables the best goodness of fitting indices are in the case of IGN+structures (better bathymetry/topography compare to GEBCO).*

8.      I suggest reorganize the conclusion section. The paragraph of uncertainty analysis should be moved to the section of 'Discussion'.

*Response: Thank you again for your comment and all the previous, we will consider this last for the new version.*
* * *
Reviewer 2

The authors present the results of a coupled Delft3D+SWAN hindcast simulation of extratropical storm Xynthia and present damage curves derived through analysis of the hindcast results in combination with insurance claims data near La Rochelle, France. Through their results, the authors demonstrate that grid/mesh resolution can impact the shape of the resulting damage curves, and that the best explanatory variables for damage are water depth and total hydrodynamic force. The authors suggest in their concluding statements that their work may have broad application to assess damage from future events along the French Atlantic coast, but subsequently provide numerous qualifiers on their work that contradict the preceding claim.

This certainly is an interesting piece of work and I believe it has a strong foundation that can be improved upon in subsequent revisions. While the technical focus is appropriate, I found the current version of the manuscript lacking in a few substantial ways. The deviation from standard practice when developing damage curves notwithstanding, the work shows promise and will be an excellent contribution to the published literature with a few substantial improvements to the analyses and manuscript.

*Dear Bret, thank you for your comments, we will certainly improve the manuscript with your comments and the other reviwers/participants on the interactive discussion forum. Down below I will answer point by point your comments.*

My general recommendations for improving the manuscript are as follows:

1)      Improve the organization of the manuscript, especially the early sections of the text. There should be a clear and distinct progression from the introduction to the methods. The methods section contains information about the study area and the storm, which would be better presented in the introductory section of the manuscript. The organization of the methods section is inconsistent and could be improved to flow more logically. For example, there is discussion of the models and model setup in multiple places of 2.1, 2.2, and 2.3. Furthermore, this section begins with (cf. 2.1) a detailed discussion of the particular storm event without first describing the storm or the models. Section 2.2 could be combined with another section in the reorganization. One may also argue that the validation results belong in the "Results" section, not in the method section. Section 2.4, albeit brief, is appropriately placed and contains helpful information. I will, however, note that use of the term "damage level x" in line 124 is somewhat inconsistent with your chosen approach and terminology. Also, that "x" is not the damage ratio but rather the value of the conditional variable (the hazard) for a specific damage ratio increment. Therefore, you likely need a subscript on P such that $P\_i(x)=\ldots$ gives the probability of experiencing hazard value "x" for damage ratio value "i" and so on. I'm sure that is what you did in the analysis, but the typesetting of Equation 2 and the corresponding text should be improved.

*We think that the best way to structure the paper is to start mentioning the economic damages and the damage functions development, instead of focusing on the storm Xynthia and the hydrodynamic simulation. Indeed it is mentioned below that we should give more importance on the damage function development than the hydrodynamic simulations. Nevertheless, due to your comment and those from reviewer 1, we decide to include another paragraph about "the common way of developing damage functions? what do the previous researchers have done? what are the main conclusions of their works?" and move some part from the conclusions to the discussion. A short sentence with reference on the introduction related with the Xynthia storm was added too*

*Related with the explanation of the damage function, we agreed that now not clear the explanation, reason why for the new version, the text will be as follows: "where P(x) is the cumulative probability of the damage ratio with values between 0 and 1, and x is the hydrodynamic variable, Φ  is the standardized normal distribution, μ is …".*

2) The analyses, while well intentioned, are not particularly robust in their presentation. For example, there is no quantitative assessment of model errors or bias in the prediction of either water levels or wave heights. Simply plotting predictions and measurements and saying the agreement is "good" does not inspire confidence, particularly when the disagreement between the two for wave heights appears to be quite substantial. As a second example, there really is not enough information provided relative to the development of the damage curves given its prominence in the title of your manuscript. So while the content of the existing manuscript is strong, it is simply short on details and could benefit from an expanded discussion in many places (a few of which are noted below).

*In the newer version we are including a goodness of fit indexes (RMSE, Pearson coefficient and RRMSE), between the observed tide and waves values and we compare these values with others values obtained in the literature.*

1)     There is a duality in the manuscript that I am having a hard time reconciling, particularly given point #2 above (lack of detail). There is a significant emphasis placed on the influence of grid resolution on the resulting damage functions. However, there is not enough supporting detail provided for these grids/meshes. Given that there is similarly a lack of detailed information regarding the development and application of the damage curves (additional comments below), this leaves the manuscript lacking in technical details as mentioned earlier. While the impact/influence of the grid resolution is noteworthy, it does not appear to be the focal point of the paper (not in the title) so I would suggest minimizing its relevance and adding much more detail to the damage curve discussion. Alternatively, if the authors would prefer not to expand the discussion of the damage curves and instead reorient the focus of the paper to one associated with the grid resolution, then consider greatly expanding details regarding the features and characteristics of those grids and perhaps modify the manuscript title accordingly.

*Yes, indeed the idea of the paper is not to focus on the mesh resolution. Instead, in the current paper we investigate three different bathymetry/topography data.resolution.*

4) I have some reservations about your analysis methodology. Not to say that it is in any way "wrong" but it does suffer from a lack of explanation (again, just my opinion). I would like to see some detailed description of the building archetypes considered in this analysis. Are all buildings considered to be of the same archetype (I assume so because there is no differentiation in the results)? Can you provide more details beyond "stone masonry" such as number of floors/heights, foundation types, age of structures, roof types/materials, etc.? Without the qualifiers, I think it would be very easy for someone to misapply your methodology.

*For this initial part of point 4 unfortunately no more information was available (such as the types of structures, number of floors and damage stages), even more the location of the claims due to the new European data protection policy were masked, making the extraction of the hydrodynamic variables an iterative process between AXA and TU-Delft.*

Also, I would like to see a better presentation of the explanatory variable (hazard) values for the damage curves. I know that you have presented them graphically in the appendix, but it would be valuable to also list the means and standard deviations (likely for only one grid) of those variables/variable groups. Finally, can you add some discussion regarding potential weaknesses of your chosen "damage ratio" approach to representing damage? There are many weaknesses with using this as a substitute for the more common "damage state" because the damage ratio does not correct for valuation based on location among other weaknesses. As another example, "insured value" is often a personal/elective choice made by the homeowner and there is bound to be substantial inconsistency in what one chooses to insure their property for. To expand a bit further, a low damage ratio value may be the result of minimal damage or a very high insured value. Therefore, the damage ratio is sensitive to two metrics, one of them choice-based, as opposed to a traditional damage state classification which, while somewhat objective, focuses only on the severity of damage to the property. My primary concern here is founded upon the fact that nearly one-half of your 423 reported claims have an assigned damage ratio <0.1 (cf. Figure 1). Finally, in a traditional damage/fragility analysis one would also consider structures with no damage. I do not recall any mention of non-damaged structures in your analysis. Therefore, the resulting damage curves may very well be biased.

***Regarding the means and the standard deviations we believe that in order to maintain an adequate extension of the paper we should not include another table since they are already displayed in the box-whisker plots. For the newer version, we add a paragraph commenting the uncertainty that the quality of the claims add on the analysis, since unfortunately cannot be assessed. As you commented, the insured value, or the economic claim can be exaggerated by the owners. About the 423 reported claims and the distribution of their damage ratio, as you mentioned, is a matter that involves the quality of the claims itself, but also the hydrodynamics conditions, that are particular for every case. In the case of La Rochelle and more noticeable for Ille du Re, there are structures at the foreshore, but usually the more dense populated areas are some hundreds of meters inland were elevation is higher and then less damages is expected. Finally, disposing of non-damaged structures indeed will add value into this kind of analysis, nevertheless we didn't dispose of this information too.***

Here are some additional comments that address specific items in the manuscript…

Line 30/Figure 1: recommend normalizing the ordinate values by the total number of claims so that you can report these in terms of their true "frequency" instead of simply counts. If not, please edit the axis title as these are not frequencies.

***Yes, the figure is changed at the new version.***

Line 38: the introduction in its current form is significantly lacking in terms of a thorough review of pertinent literature on damage functions derived from coastal hazard models (e.g., Masoomi et al., 2019 and many others), lacks an orientation to the study area, and does not thoroughly describe the storm event. I would recommend adding:

-much more background on relevant literature

- a detailed description of study area with location map, exposure/vulnerability to extreme events, hydrodynamic setting, etc.

- more information on the history and characteristics of Xynthia

Masoomi, H., van de Lindt, J.W., Do, T.Q., Webb, B.M. 2019. Combined wind-wave-surge hurricane-induced damage prediction for buildings. Journal of Structural Engineering 145(1).

*Yes, this reference and others, both together a better explanation for the damage functions and Xynthia storm is added.*

Line 41: minor comment but use consistent typesetting of "Delft3D" throughout the document.

*Yes, thank you is amended.*

Line 48/Figure 2: any reason why there is a font change in this graphic? Was that intentional?

*Fonts on the figures are checked and corrected*

Lines 49-69: I find it odd that you are interjecting more literature review here as opposed to providing it earlier in the document.

*As commented before more references are added in the beginning of the document*

Line 63: missing comma… "… storm characteristics, a regional model…"

*Yes, Thank you is amended.*

Line 65: can a resolution of 80 meters accurately capture terrain features and individual homes?

*Homes and small terrain features are not included in the grid resolution but flood walls were included by means of thin weirs in the model*

Line 69/Figure 3: There is not enough contrast in this image to make out the details. The date/time codes for every storm report make the figure unnecessarily busy.

*Yes, some labels were removed and an additional image was included.*

Line 73: what data sources did you use for land cover / land use to assign friction coefficients? (nb. Ignore this comment, I see the answer on line 99).

*See comments on line 99.*

Lines 75-79/Table 1: how do these relate to the 80-meter resolution mentioned previously?

*As commented above, the intention is to compare different bathymetry/topography information data. Resolution of this information is variable from 500m (GEBCO) to 5m (IGN topography). The mesh of 80m can produce different results depending on the type of the information.*

*Our analysis consist in three domains because we implemented the model in domain decomposition, but this is not related to our analysis of the data resolution. For each DEM (GEBCO, IGN, IGN +Structures) we implemented that DEM in all of our domains.*

Line 83: "… spatial resolution and temporal every 3hrs." awkward phrasing

*Ok, the sentence has been changed*

Line 99: use of a constant Manning's "n" value for the entire grid is a significant technical weakness in this study. While it "may" be appropriate for some open water conditions, it is certainly not reflective of the terrain where the subject structures were likely found. Could you please provide a justification and suitable citation to support the use of a constant friction factor? I have read numerous papers in the past ten years that point to the importance of accurate representation of terrain roughness through the assignment of proper friction coefficients.

*We do agree that the roughness friction coefficients have an influence on the hydrodynamic results, particularly in places of shallow waters, but in order to simplify the analysis as was done in Tomiczek T., 2017, we decide to keep it constant. A small paragraph on the discussion is added regarded this topic.*

Line 105: what is SHOM?

*A web link is added. Is an institute in charge of the monitoring of the north Atlantic bouys in front of the French and Spanish coast.*

Line 105: "… during the whole simulation is good (Figure 4)." Is good relative to what? There is no quantitative basis for this statement.

*A table with the goodness of fit indicators is added. Also a small explanation on the reason by which the significant wave height is compared to the swell height is done.*

Line 110/Figure 4: Can you please explain the spurious oscillation at the beginning of your simulation results? I incorrectly assumed that this was the surge event, but it appears to be a numerical instability associated with model spinup.

*Since at the time of the beginning of the simulation the water levels in the whole domain are unknown, it is common to set as 0 m.a.s.l the water elevation as initial condition. This produce the need to extend the simulation and start it before the day from which we want to make the analysis. How long is this period (or spin-up time) depends on the size of the domain and time interval of the simulation. In our case we start the simulation from the 20 of February of 2010 guarantying enough time for the model to adjust the tide values to the real ones.*

Line 115/Figure 5: there is absolutely no assessment or narrative to accompany these results. Since wave properties are highlighted as one of your preferred explanatory variables in the damage analysis, and since wave processes contribute to coastal flooding (your other explanatory variable), can you provide some commentary on the disagreement between the modeled and measured waves? Why are the observations listed as "swell height"? Are you comparing two different wave statistics in this figure (i.e., swell height and significant wave height)?

*Yes, thank you for this observation. An explanation is added both together with the goodness of fit indices for the waves and tides are added.*

Line 121: "… specifically, relates the…" typo and awkward phrasing

*Ok, sentence has been changed.*

Line 127: missing commas after "paper" and "way"

*Ok, sentence has been changed.*

Lines 132-138: Damage curves are often given for different levels/magnitudes of damage. Here it appears that you are integrating the damage results across all of the discrete damage ratio increments. Was this a specific choice/preference, an artifact of your damage indicator scheme, or something else entirely? Was there no interest in disaggregating the damage data as is traditionally done in these types of analyses? For example, developing unique damage curves for different damage states?

*Unfortunately we do not have enough data to disaggregate.*

Line 133: I don't think "Box-Whisker" is capitalized, not proper nouns

*Thank you for the observation.*

Line 136: "Damage" should not be capitalized

*Thank you for the observation.*

Line 142: "Where" should not be capitalized since you are using the equations as the subject of your sentence.

*Thank you for the observation.*

Line 145/Figure 6: I find these figures to be less helpful than I had hoped. The size (small) and contouring scheme do not allow for much interpretation of the results.

*Figure 6 is changed modifying the colours, the legend and making a zoom over the simulation domain.*

Line 150: "Similarly" -> "Similar"

*Thank you for the observation*

Line 152/Figure 7: at the scale provided it is difficult to discern details in these figures. Also, there is no explanation of the symbols in these figures.

*Thank you for the observation figure is change*

Line 157: "related with the" -> "related to the"

*Thank you for the observation*

Line 160/Table 2: For explanatory values (cf Table 2) used in Eq 2, how were means and standard deviations evaluated for combinations of variables that do not necessarily vary consistently in time? In other words, did you estimate the time-variation of each variable group/combination and then take the mean and standard deviation of the entire time-series? Or, did you evaluate the mean/stdev of each individual parameter and then form the variable groups?

*For every single claim at the Ille du Re and surroundings the maximum hydrodynamic variables were extracted at every single location. In this way the time series of the simulations are restricted for the maximum values that reach every single variable. A short line explaining this is added in the damage curve section.*

Line 164: typesetting of "hsig" -> "Hsig"

*Thank you for the observation.*

Line 207: delete comma after point

*Thank you for the observation.*

Line 215: "… as thin or concrete structures like flood walls at typically only a few 10's of centimeters thick, and so do not appear in digital elevation models." Awkward phrasing.

*Yes, the phrase is adjusted.*

Lines 215-220: what about errors/uncertainty in your model predictions?

*Thank you for the comment. It was added along the document at the meteorological and damage functions some paragraphs about the uncertainty coming from the hydrodynamic modelling and from the claims data itself in the document.*

I commend the authors on a very strong first draft of what I'm sure was a very challenging manuscript to prepare. The authors are absolutely on track towards having a very strong publication that will productively add to the body of literature on damage to coastal structures during extreme events.

Sincerely,

Bret Webb

*We certainly acknowledge your comments on the document, thank you for reading the document and give your feedback. We hope the present version is more accurate.*

**Andres et al.,**

Editor (Sven Fuchs)

Dear colleagues,

thank you very much for submitting your interesting and timely manuscript for consideration in NHESS. During the public discussion, we received three very comprehensive comments on your work, two from the reviewers and one community comment. All three raised some issues, which you have already extensively commented in your author responses.

As always, we can also detect a certain bias on the individual concern of these three comments, as the individual disciplinary background of the colleagues is obviously a bit different. As such it is clear that these comments do not overlap 1:1.

Based on the comments and your answers to these I decided to give you the necessary time for a major revision of your work. It may be – from my side – an advice that you clearly make a statement in the very early introduction that your work is targeted at developing vulnerability curves based on the results of hazard modelling (and as such, the hazard modelling may be not as precise as for contributions solely targeted at back-calculation of a hazard event itself).

In other words, you may wish to focus specifically on the comments of referee 1 on the background of loss functions and the tricky issue with the assumed standard normal distribution (you may wish to refer also to the "Short Communication" on a Beta model explicitly focusing on this issue, but of course for another hazard type, see https://doi.org/10.1016/j.envsoft.2019.03.026). Moreover, also referee #2 provided comprehensive advice on how to streamline the entire manuscript (points 1-4 specifically), in particular with respect to the missing details of modelling. You may also think about putting some of the material in an Appendix (or more) so that the overall string of arguments may become clearer. A revised introduction will certainly also pick-up thoughts of the community comment of Xavier Bertin.

I wish you good success with further developing the manuscript and I also kindly ask you to contact the Copernicus team (via replying to the e-mail) at your earliest convenience if you need more time for your revisions than foreseen by the system – this has to be done manually, but is in case of your work no problem, of course.

Kind regards,

Sven Fuchs (Editor NHESS)

*Dear editor, we have made extensively changes in the manuscript including your comments above mentioned. We hope this time the manuscript reach the appropriate correctness for publication.*

*Thank you,*

*Andres et al.*

---

## Author Response (AR2)

Dear colleagues,

meanwhile I received the referee opinions of this second round of review.

As you can see, referee #1 who was already involved in the first round has some minor comments, mainly based on final clarification of the manuscript content and some linguistic remarks. I have the feeling that you will be able to address these issues during a final minor revision.

Referee #3 who was not involved in the first round raised some more general questions. The overall debate in physical vulnerability assessment is whether this is based on deterministic/empirical loss functions or on more detailed fragility curves based on an individual assessment of structures. The latter, as far as I understood also from the first review round, remarks to referee #2 (Bret Webb), was not possible due to data protection and other restrictions associated with the exposure data you used. This is an overall challenge in vulnerability assessment, and has also been experienced by many other fellows working with insurance claims and/or aggregated loss data (to give an example, if you only know the degree of loss composed from the building value and the loss height, but have not additional information on the building type, this is the way to proceed). So my suggestion with respect to the comments of referee #2, though they are very valuable, is to add a paragraph in the final section on the limits of your approach. This could also been done by extending the last paragraph of Section 4 (see as an example Fuchs et al., 2019, section 3).

Based on the options of both of the referees and my own judgement I consider your work as timely and publishable, however, I kindly ask you to re-consider the issues raised above in terms of a minor revision before I will accept your very interesting piece of work.

I am looking forward to receiving your revised manuscript.

Kind regards,

Sven Fuchs (Editor NHESS)

**Response:**

*Dear Sven.*

*Yes, we added a paragraph at the end of section 4 explaining the limitations of our approach regarding the available information.*

*We would like to thank you for the help over these months.*

*Best,*

*Andres*

Reviewer3

The paper develops damage curves using the insurance claim data. Although the work should be of great interest, I expect rigorous efforts from the authors before it can be considered for publication in NHESS. My suggestions are as follows:

Please use what type of structures you are considering in the study. Since the classification of structures is quite risky due to the variation in the attributes, classifying data into a bin requires a very high level of understanding. Please mention how did you define classes of structures, also please explain the attributes of structures and the reason behind homogenization. Insurance claims will be randomly done for all structures and creating damage curves for structures does not guarantee representativeness. Rather, it would be imperative to disaggregate the data into several classes and define fragility functions or vulnerability functions. An introduction of vulnerability/fragility is also welcomed.

I suggest the authors use fragility curves rather than damage curves. At some point, these terms differ at least when we go for the classical definition of fragility. Usually, if you say damage curve, it becomes rather deterministic and may even miss the accumulating nature. What I mean is, higher damage by default carries the lower one.

Since flooding damage is not confined to Europe only, please include a global literature review on empirical fragility/vulnerability models such as:

- From flood risk mapping toward reducing vulnerability: the case of Addis Ababa by De Risi et al.

- An analysis of physical vulnerability to flash floods in the small mountainous watershed of Aceh Besar Regency, Aceh province, Indonesia by Azmeri and Isa

- Catchment-scale flood hazard mapping and flood vulnerability analysis of residential buildings: The case of Khando River in eastern Nepal by Thapa et al.

- Multi-hazard vulnerability of structures and lifelines due to the 2015 Gorkha earthquake and 2017 central Nepal flash flood by Gautam and Dong

Moreover, a comprehensive literature review can be found in the paper by Fuchs et al.: Recent advances in vulnerability assessment for the built environment exposed to torrential hazards: challenges and the way forward.

The authors should present a comprehensive discussion on why the least square approach is used. For me, the maximum likelihood estimation is also impressive due to several merits. A more detailed explanation of statistical modeling and selection of intensity measures is to be provided in the manuscript.

How did you estimate the damage level of each structure and on what basis did you estimate the damage level or damage factor?

**Response:**

*Dear reviewer 3.*

***Thank you for your comment. The reason why we decided to use the definition of damage curves instead of fragility curves is that we did not have more information than the damage ratio (the ratio of claim value to total insured value). In the previous round of discussion (that you unfortunately missed) this question was addressed and in the reply to the reviewers, was explained that unfortunately due to the new European data protection policy (GDPR - General Data Protection Regulation), the AXA insurance company cannot share more information with us than currently already listed in the paper.***

***Nevertheless, at the end of the section 4 one more paragraph was added mentioning this limitation on the paper as the editor recommended based on your comments.***

*Regarding the question related to the parameters adjustment method, in the present paper we use the L-moments method (included in the package lmomco in R) which is a common method to fit parameter distributions, and we mentioned other methods exist, like the least square method, or as you are now commenting, the maximum likelihood method for this task. It is true that the selection of the method can influence the results, but the scope of the present paper is not to delve into the statistical methods for damage curve development, but to research on the variables that correlate most to the damage ratio due to storm Xynthia. Indeed, a previous second reviewer commented on how other different distribution functions would affect the results, and as a complementary analysis, two newer distribution functions were added (and the analysis was repeated), showing that the water depth (d) and the total force are still the most representative variables to develop the damage curves, and more probably this result will not change by using a different parameter fitting method. Nonetheless, at the end of section 3 a small sentence was added regarding this topic.*

Bret Webb (Reviwer 2)

The authors have done a good job addressing my comments, questions, and requests. I have a few very minor comments/questions below, but I don't think these warrant any sort of substantial revision of the manuscript.

I'm not sure that the authors improved the description of Xynthia as was suggested in my comments. In fact, there appears to be less text describing the storm event now. I still think it would be worthwhile to have a more thorough description of the event and its impacts to the coast and infrastructure.

Now that you have clarified the nature of the swell wave height measurements used in Figure 5, did you consider exporting the corresponding swell wave height predictions from SWAN instead of simply using the significant wave height results? I only ask because the model-data comparison may be much more robust than your analysis indicates.

On the figure of the left will not improve on the right will improve

There are some minor spelling, grammatical, and typesetting issues throughout the manuscript that I'm sure the publishing/layout staff will find so I won't list them here.

**Response:**

*Dear Bret,*

*Thank you for your comments.*

*In the previous version we increased the description of Xynthia storm, but we re-structured the paper, therefore maybe there is a feeling of less text. Nevertheless, now a new paragraph is added in section 2.1.2 describing the storm and the damages/casualties for the event a bit more.*

*It is a good point about the Swell, unfortunately since in figure 5 left, SWAN significant wave height is underestimating the Swell and Figure 5 right is overestimating (comparing hsig and swell), therefore this will not improve the analysis on average.*

*For this final version, co-author Jeremy Bricker who is a native English speaker has reviewed the English.*

*Thank you all.,*

*Manuel Andres Diaz Loaiza et al.*